# Efficient Deep Approximation of GMMs

**Shirin Jalali, Carl Nuzman, Iraj Saniee**
Bell Labs, Nokia
600-700 Mountain Avenue
Murray Hill, NJ 07974
{shirin.jalali,carl.nuzman,iraj.saniee}@nokia-bell-labs.com

## Abstract

The universal approximation theorem states that any regular function can be approximated closely using a single hidden layer neural network. Some recent work has shown that, for some special functions, the number of nodes in such an approximation could be exponentially reduced with multi-layer neural networks. In this work, we extend this idea to a rich class of functions, namely the discriminant functions that arise in optimal Bayesian classification of Gaussian mixture models (GMMs) in $\mathbb{R}^n$. We show that such functions can be approximated with arbitrary precision using $O(n)$ nodes in a neural network with two hidden layers (deep neural network), while in contrast, a neural network with a single hidden layer (shallow neural network) would require at least $O(\exp(n))$ nodes or exponentially large coefficients. Given the universality of the Gaussian distribution in the feature spaces of data, e.g., in speech, image and text, our results shed light on the observed efficiency of deep neural networks in practical classification problems.

## 1   Introduction

There is a rapidly growing literature which demonstrates the effectiveness of deep neural networks in classification problems that arise in practice; e.g., in audio, image or text classification. The universal approximation theorem, UAT, see [1, 2, 3, 4], states that any regular function, which for example separates in the (high dimensional) feature space a collection of points corresponding to images of dogs from those of cats, can be approximated by a neural network. But UAT is proven for shallow, i.e., single hidden-layer, neural networks and in fact the number of nodes needed may be exponentially or super exponentially large in the ambient dimension of feature space. Yet, practical deep neural networks are able to solve such classification problems effectively and efficiently, i.e., using what amounts to a small number of nodes in terms of the size of the feature space of the data. There is no theory yet as to why deep neural networks (DNNs from here on) are as effective and efficient in practice as they evidently are. There are essentially two possibilities for this observed outcome: 1) DNNs are *always* significantly more efficient in terms of the number of nodes used for approximation of *any* relevant function than shallow networks, or 2) DNNs are particularly suited to discriminant functions that arise *in practice*, e.g., those that separate in the feature space of images, points representing dogs from points representing cats. If the latter proposition is true, then the observed efficiency of DNNs is essentially due to the special form of the discriminant functions encountered in practice and not to the universal efficiency of DNNs, which the former proposition would imply.

The first alternative proposed above is a general question about function approximation given neural networks as the collection of basis functions. As of today, there are no general results that show DNNs (those with two or more hidden layers) require fundamentally fewer nodes for approximation of general functions than shallow neural networks (or SNNs from here on, i.e., those with a single hidden layer). In this paper, we focus on the second alternative and provide an answer in the affirmative; that

indeed many discriminant functions that arise in practice are such that DNNs require significantly, i.e., logarithmically, fewer nodes for their approximation than SNNs. To formalize what may constitute discriminant functions that arise *in practice*, we focus on a versatile class of distributions often used to model real-life distributions, namely Gaussian mixture model (GMM for short). GMMs have been shown to be good models for audio, speech, image and text processing in the past decades, e.g., see [5, 6, 7, 8, 9].

## 1.1 Background

The universal approximation theorem [1, 2, 3, 4] states that shallow neural networks (SNNs) can approximate regular functions to any required accuracy, albeit potentially with an exponentially large number of nodes. Can this number be reduced significantly, e.g., logarithmically, by deep neural networks? As indicated above, there is no such result as of yet and there is scant literature that even discusses this question. Some evidence exists that DNNs may in fact not be efficient in theory, see [10]. On the other hand, some special functions have been constructed for which DNNs achieve significant and even logarithmic reduction in the number of nodes compared to SNNs, e.g., see [11, 12, 13] for a special radial function, functions of the form $f(\mathbf{x}, \mathbf{x}') = g(\langle \mathbf{x}, \mathbf{x}' \rangle)$ ($f : \mathcal{S}^{n-1} \times \mathcal{S}^{n-1} \to \mathbb{R}$ and $g : [-1, 1] \to \mathbb{R}$), and polynomials, respectively. However, the functions considered in these references are typically very special and have little demonstrated basis in practice. Perhaps the most illustrative cases are the high degree polynomials discussed in [13], but the logarithmic reduction in the number of nodes due to depth of the DNNs demonstrated in this work occurs only for very high degrees of polynomials in the feature coordinate size.

In this work we are motivated by model universality considerations. What models of data are typical and what resulting discriminant functions do we typically need to approximate in practice? With a plausible model, we can determine if the resulting discriminant function(s) can be approximated efficiently by deep networks. To this end, we focus on data with Gaussian feature distributions, which provide a practical model for many types of data, especially when the feature space is sufficiently concentrated, e.g. after a number of projections to lower-dimensional spaces, e.g., see [14].

Our overall framework is based on the following set of definitions and demonstrations that we describe in detail in the following sections. Section 1.2 defines an $L$-layer neural network. Section 1.3 reviews the problem of optimal classification of a collection of high-dimensional GMMs. The classifier function for GMMs is readily seen to be the maximum of multiple discriminant functions consisting of sums of exponentials of quadratic functions in dimension $n$. Section 2 establishes connections between approximating the defined discriminant functions with the described classification problem. Section 3 demonstrates that DNNs can approximate general $n$-dimensional GMM discriminant functions using $O(n)$ nodes. In Section 4, we show that, in contrast to DNNs, SNNs need either an exponential (in $n$) number of nodes, or exponentially large coefficients to approximate discriminant functions of GMMs. In Section 5, we show sufficiency of an exponential number of nodes by studying an SNN where the weights in the first layer are drawn from a random distribution.

*Notation.* Throughput the paper, bold letters, such as $\mathbf{x}$ and $\mathbf{y}$, refer to vectors. Sets are denoted by calligraphic letters, such as $\mathcal{X}$ and $\mathcal{Y}$. For a discrete set $\mathcal{X}$, $|\mathcal{X}|$ denotes its cardinality. $\mathbf{0}_n$ denotes the all-zero vector in $\mathbb{R}^n$. $I_n$ denotes the $n$-dimensional identity matrix. For $\mathbf{x} \in \mathbb{R}^n$, $\|\mathbf{x}\|^2 = \sum_{i=1}^n x_i^2$.

## 1.2 $L$-layer neural networks and the activation function $\sigma$

*L-layer Neural Network.* Consider a fully-connected neural network with $L$ hidden layers. We refer to a network with $L = 1$ hidden layer as an SNN and to a network with $L > 1$ hidden layers as DNN. Let $\mathbf{x} \in \mathbb{R}^n$ denote the input vector. The function generated by an $L$-layer neural network, $f : \mathbb{R}^n \to \mathbb{R}^c$, can be represented as a composition of affine functions and the non-linear function $\sigma$, as follows $f(\mathbf{x}) = \sigma \circ T^{[L+1]} \circ \sigma \circ T^{[L]} \ldots \sigma \circ T^{[1]}(\mathbf{x})$. Here, $T^{[\ell]} : \mathbb{R}^{n_{\ell-1}} \to \mathbb{R}^{n_\ell}$ denotes the affine mapping applied at layer $\ell$, represented by linear transformation $W^{[\ell]} \in \mathbb{R}^{n_\ell \times n_{\ell-1}}$ and translation $\mathbf{b}^{[\ell]}$. Moreover, $\sigma : \mathbb{R} \to \mathbb{R}$ denotes the non-linear function that is applied element-wise. In this definition, $n_\ell, \ell = 1, \ldots, L$, denotes the number of hidden nodes in layer $\ell$. To make the notation consistent, for $\ell = 0$, let $n_0 = n$, the dimension of the input data, and for $\ell = L + 1$, $n_{L+1} = c$, the number of classes. We will occasionally use the notations $dnn$ and $snn$ to signify that the cases of $L > 1$ and $L = 1$, respectively. In a classification task, the index of the highest value in the output tuple determines the optimal class for $\mathbf{x}$.

*Non-linear activation function $\sigma$.* As discussed later, for the two-layer construction in Section 3, we require some regularity assumptions on the activation function $\sigma$, which are met by typical smooth DNN activation functions such as the sigmoid function. In Section 7, we indicate how to refine the proofs to accommodate the popular and simple ReLU activation function. The proof of the inefficiency of SNNs in Section 4 applies to a very general class of activation functions.

### 1.3 GMMs and their optimal classification functions

Consider the problem of classifying points generated by a mixture of Gaussian distributions. Assume that there are $c$ classes and samples of each class are drawn from a mixture of Gaussian distributions. Assume that there are overall $k$ different Gaussian distributions to draw from. For $j \in \{1, \ldots, k\}$, let $\boldsymbol{\mu}_j \in \mathbb{R}^n$ and $\Sigma_j \in \mathbb{R}^{n \times n}$ denote the mean and the covariance matrix of Gaussian distribution $j$. Each Gaussian distribution is assigned uniquely to one of the $c$ classes. Assume that the assignment of the Gaussian clouds to the classes is represented by sets $\mathcal{T}_1, \ldots, \mathcal{T}_c$, which form a partition of $\{1, \ldots, k\}$. (That is, $\mathcal{T}_i \cap \mathcal{T}_j = \emptyset$, for $i \neq j$, and $\cup_{i=1}^c \mathcal{T}_i = \{1, \ldots, k\}$.) Set $\mathcal{T}_i$ represents the indices of the Gaussian distributions corresponding to Class $i$. Let $\phi_i$, $i = 1, \ldots, c$, denote the probability that a data point belongs to Class $i$. Finally, for Class $i$, let $w_j$, $j \in \mathcal{T}_i$, denote the probability that within Class $i$, the data comes from Gaussian distribution $j$. Hence, for $i = 1, \ldots, n$, $\sum_{j \in \mathcal{T}_i} w_j = 1$. Under the described model, and with a slight abuse of notation, the data is distributed as $\sum_{i=1}^c \phi_i \sum_{j \in \mathcal{T}_i} w_j \mathcal{N}(\boldsymbol{\mu}_j, \Sigma_j)$. Conditioned on being in Class $i$, the data points are drawn from a mixture of $|\mathcal{T}_i|$ Gaussian distributions as $\sum_{j \in \mathcal{T}_i} w_j \mathcal{N}(\boldsymbol{\mu}_j, \Sigma_j)$. For $j = 1, \ldots, k$, let $\pi_j : \mathbb{R}^n \to \mathbb{R}$ denote the probability density function (pdf) of the Gaussian distribution with mean $\boldsymbol{\mu}_j$ and covariance matrix $\Sigma_j$.

An optimal Bayesian classifier[1] $\mathcal{C}^* : \mathbb{R}^n \to \{1, \ldots, c\}$ for these $c$ GMMs maximizes the probability of membership across all classes. For Class $i$, define the $i$-th discriminant function $d_i : \mathbb{R}^n \to \mathbb{R}$, as

$$d_i(\mathbf{x}) \triangleq \phi_i \sum_{j \in \mathcal{T}_i} w_j \gamma_j \exp(-g_j(\mathbf{x})), \tag{1}$$

where $g_j(\mathbf{x}) \triangleq \frac{1}{2}(\mathbf{x} - \mu_j)^T \Sigma_j^{-1} (\mathbf{x} - \mu_j)$ and $\gamma_j \triangleq (2\pi)^{-\frac{n}{2}} |\Sigma_j|^{-\frac{1}{2}}$. Using this definition, the optimal classifier $\mathcal{C}^*$ can be characterized as

$$\mathcal{C}^*(\mathbf{x}) = \arg\max_{i \in \{1, \ldots, c\}} d_i(\mathbf{x}). \tag{2}$$

## 2 Connection between classification and approximation

The main result of this paper is that the discriminant functions described in (1), required for computing optimal classification function $\mathcal{C}^*(\mathbf{x})$, can be approximated accurately by a relatively small neural network with *two* hidden layers, but that accurate approximation with a *single* hidden layer network is only possible if either the number of the nodes or the magnitudes of the coefficients are exponentially large in $n$. Before stating our main results, in this section, we establish a connection between the accuracy in approximating the discriminant functions of a classifier and the error performance of a classifier that employs these approximations.

Given a non-negative function $d(\mathbf{x})$, $d : \mathbb{R}^n \to \mathbb{R}$, and threshold $t > 0$, let $\mathcal{S}_{d,t}$ denote the superlevel set of $d(\mathbf{x})$ defined as

$$\mathcal{S}_{d,t} \triangleq \{\mathbf{x} \in \mathbb{R}^n : d(\mathbf{x}) \geq t\}.$$

**Definition 1** *A function $\hat{d} : \mathbb{R}^n \to \mathbb{R}$ is a $(\delta, q)$-approximation of a non-negative function $d : \mathbb{R}^n \to \mathbb{R}$ under a pdf $p$, if there is a threshold $t$, such that $\mathrm{P}_p[\mathcal{S}_{d,t}] \geq 1 - q$, and*

$$|\hat{d}(\mathbf{x}) - d(\mathbf{x})| \leq \delta d(\mathbf{x}), \qquad \mathbf{x} \in \mathcal{S}_{d,t} \tag{3}$$

$$0 \leq \hat{d}(\mathbf{x}) \leq (1 + \delta)t, \qquad \mathbf{x} \notin \mathcal{S}_{d,t}. \tag{4}$$

*Let $t_{\hat{d},\delta,q}$ denote the corresponding threshold. If there are multiple such thresholds, let $t_{\hat{d},\delta,q}$ denote the infimum of all such thresholds.*

In this definition, $\hat{d}$ closely approximates $d$ in a relative sense, wherever $d(\mathbf{x})$ exceeds threshold $t$. The function $\hat{d}$ is small (in an absolute sense), when $d(\mathbf{x})$ is small, an event that occurs with low probability under $p$. Although $p$ and $d$ need not be related in this definition, we will typically use it in cases where $d$ is just a scaled version of $p$.

Given two equiprobable classes with pdf functions $p_1$ and $p_2$, the optimal Bayesian classifier chooses class 1, if $p_1(\mathbf{x}) > p_2(\mathbf{x})$, and class 2 otherwise. Let $e_{21,\text{opt}} = \mathrm{P}_1[p_2(\mathbf{x}) > p_1(\mathbf{x})]$ denote the probability of incorrectly deciding class 2, when the true distribution is class 1. If we classify using *approximate* pdfs with relative errors bounded by $\alpha \geq 1$, then the probability of error increases to $e_{21,\text{opt}}[\alpha] := \mathrm{P}_1[p_2(\mathbf{x}) > p_1(\mathbf{x})/\alpha]$. Under appropriate regularity conditions, $e_{21,\text{opt}}[\alpha]$ approaches $e_{21,\text{opt}}$, as $\alpha$ converges to 1. Lemma 1 below shows that $(\delta, q)$-approximations of $p_1$ and $p_2$ enable us to approach $e_{21,\text{opt}}$, by taking $\delta$ and $q$ sufficiently small.

**Lemma 1** *Given pdfs $p_1$ and $p_2$, let $\hat{d}_1$ and $\hat{d}_2$ denote $(\delta, q)$-approximations of discriminant functions $d_1 = p_1$ and $d_2 = p_2$ under distributions $p_1$ and $p_2$, respectively. Define $t_i$, $i = 1, 2$, as $t_i \triangleq t_{\hat{d}_i, \delta, q}$. Consider a classifier that declares class 1 when $\hat{d}_1(X) > \hat{d}_2(X)$ and class 2 otherwise. Then, the probability of error of this classifier, under distribution $p_1$, is bounded by*

$$e_{21} \leq e_{21,\text{opt}}\left[\tfrac{1+\delta}{1-\delta}\right] + q + \mathrm{P}_1(\mathcal{S}^c_{d_1, (1+\delta)t_2/(1-\delta)}),$$

*where $\mathrm{P}_1(\mathcal{E})$ measures the probability of event $\mathcal{E}$ under $p_1$.*

The proof of Lemma 1 is presented in Section 1 of the supplementary material (SM).

Note that as $q$ converges to zero, both $t_1$ and $t_2$ converge to zero as well. Therefore, letting $q$ converge to zero ensures that $\mathrm{P}_1((1+\delta)t_2 \geq (1-\delta)d_1(\mathbf{x}))$ also converges to zero. One way to construct a nearly optimal classifier for two distributions is thus to independently build a $(\delta, q)$-approximation for each distribution, and then define the classifier based on maximum of the two functions.

With this motivation, in the rest of the paper, we focus on approximating the discriminant functions defined earlier for classifying GMMs. In the next section, we show that using a two hidden-layer neural network, we can construct a $(\delta, q)$-approximation $\hat{d}$ of the discriminant function of a GMM, see (1), with input dimension $n$, with $O(n)$ nodes, for any $\delta > 0$ and any $q > 0$.

In the subsequent section, we show by contrast that even for the simplest GMM consisting of a single Gaussian distribution, even a weaker approximation that bounds the *expected* $\ell_2$ error cannot be achieved by a single hidden-layer network, unless the (neural) network has either exponentially many nodes or exponentially large coefficients. The weaker definition of approximation that we will use in the converse result is the following.

**Definition 2** *A function $\hat{d} : \mathbb{R}^n \to \mathbb{R}$ is an $\epsilon$-relative $\ell_2$ approximation for a function $d : \mathbb{R}^n \to \mathbb{R}$ under pdf $p$, if*

$$\mathrm{E}_p[(\hat{d}(\mathbf{x}) - d(\mathbf{x}))^2] \leq \epsilon\, \mathrm{E}_p[(d(\mathbf{x}))^2].$$

The following lemma shows that if approximation under this weaker notion is impossible, then it is also impossible under the stronger $(\delta, q)$ notion.

**Lemma 2** *If $\hat{d}$ is a $(\delta, q)$-approximation of a distribution $d$ under distribution $p$, then it is also an $\epsilon$-relative $\ell_2$ approximation of $d$, with parameter $\epsilon = \delta^2 + \frac{(1+\delta)^2 q}{1-q}$.*

Proof of Lemma 2 is presented in Section 2 of the SM.

## 3 Sufficiency of two hidden-layer NN with $O(n)$ nodes

We are interested in approximating the discriminant functions corresponding to optimal classification of GMMs, as defined in Section 1.3. In this section, we consider a generic function for (1) as

$$d(\mathbf{x}) = \sum_{j=1}^{J} \beta_j \exp\left(-g_j(\mathbf{x})\right), \tag{5}$$

where $g_j(\mathbf{x}) = \frac{1}{2}(\mathbf{x} - \mu_j)^T \Sigma_j^{-1}(\mathbf{x} - \mu_j)$ and $\beta_j = \phi w_j \gamma_j$ with fixed prior $\phi$, conditional probabilities $w_j$, and $\gamma_j = (2\pi)^{-\frac{n}{2}} |\Sigma_j|^{-\frac{1}{2}}$.

We first observe that the function $g_j(\mathbf{x})$ is a general quadratic form in $\mathbb{R}^n$ and thus consists of the sum of $O(n^2)$ product terms of the form $x_i x_j$. Since each such product term can be approximated via four $\sigma$ functions (see [15]), $g_j(\mathbf{x})$ can be approximated arbitrarily well using $O(n^2)$ nodes. We can however reduce the number of nodes further by applying the affine transformation $\mathbf{y}_j = \Sigma_j^{-1/2}(\mathbf{x} - \boldsymbol{\mu}_j)$ in the first layer, so that $g_j(\mathbf{x})$ is simply $\|\mathbf{y}_j\|^2 = \sum_{i=1}^n y_{j,i}^2$, i.e., it consists of $n$ quadratic terms, and can in turn can be approximated via $O(n)$ nodes. This $O(n^2)$ to $O(n)$ reduction in the number of nodes is specific to quadratic polynomials which are generic exponents of GMMs and their discriminant functions and we will take advantage of this reduction in our proofs.

In order to prove the main result of this section, we rely on the following regularity assumptions on the activation function $\sigma(x)$. All of the assumptions are satisfied by the sigmoid function $\sigma(x) = \frac{1}{1+e^{-x}}$, for example.

**Assumption 1 (Curvature)** *There is a point $\tau \in \mathbb{R}$, and parameters $r$ and $M$, such that $\sigma^{(2)}(\tau) > 0$, such that $\sigma^{(3)}(x)$ exists and is bounded, $|\sigma^{(3)}(x)| \leq M$, in the neighborhood $\tau - r \leq x \leq \tau + r$.*

**Assumption 2 (Monotonicity)** *The symmetric function $\sigma(x + \tau) + \sigma(-x + \tau)$ is monotonically increasing for $x \geq 0$, with $\tau$ as defined in Assumption 1.*

**Assumption 3 (Exponential Decay)** *There is $\eta > 0$ such that $|\sigma(x)| \leq \exp(\eta x)$ and $|1 - \sigma(x)| \leq \exp(-\eta x)$.*

Assumptions 1 and 2 can be used to construct an approximation of $x^2$ using $O(1)$ nodes for each such term. They are satisfied for example by common activation functions such as the sigmoid and $\tanh$ functions. Assumption 3 relied upon to construct an approximation of $\exp(x)$ in the second hidden layer, with $O(n)$ nodes in each of the $J$ subnetworks. This assumption is met by the indicator function $u(x) = 1\{x > 0\}$, and any number of activation functions that are smoother versions of $u(x)$, including piecewise linear approximations of $u(x)$ constructed with ReLU, and the sigmoid function.

The following is our main positive result about the ability to efficiently approximate GMM discriminant functions with two-layer neural networks.

**Theorem 1** *Consider a GMM with discriminant function $d : \mathbb{R}^n \to \mathbb{R}^+$ of the form (5), consisting of Gaussian pdfs with bounded covariance matrices. Let the activation function $\sigma : \mathbb{R} \to \mathbb{R}$ satisfy Assumptions 1, 2, and 3 . Then for any given $\delta > 0$ and any $q \in (0, 1)$, there exists a two-hidden-layer neural network consisting of $M = O(n)$ instances of the activation function $\sigma$ and weights growing as $O(n^5)$, such that its output function $\hat{d}$ is a $(\delta, q)$-approximation of $d$, under the distribution of the GMM.*

The detailed proof of Theorem 1 is presented in Section 5 of the SM. The proof relies on several lemmas that are stated and proved in Section 4 of the SM.

**Remark 1** *Applying Theorem 1 to a collection of $c$ GMMs gives rise to a DNN with $O(n)$ nodes that approximates the optimal classifier of these GMMs via (2).*

**Remark 2** *The construction of an $O(n)$-node approximation of the GMM discriminant function assumes that the eigenvalues of the covariance matrices are bounded from above and also bounded away from zero, by constants independent of $n$.*

To prove Theorem 1, given $d(\mathbf{x}) = \sum_{j=1}^J \beta_j \exp(-g_j(\mathbf{x}))$, we build a neural net consisting of $J$ sub-networks, with sub-network $j$ approximating $\beta_j \exp(-g_j(\mathbf{x}))$. For convenience, denote by $c_j(\mathbf{x}) = \beta_j \exp(-g_j(\mathbf{x}))$, the desired output of the $j$-th subnetwork. The $J$ subnetwork function approximations $\hat{c}_j(\mathbf{x})$, $j = 1, \ldots, J$, are summed up to get the final output $\hat{d}(\mathbf{x}) = \sum_j \hat{c}_j(\mathbf{x})$.

**Lemma 3** *Given $\delta > 0$, $q > 0$, and the GMM discriminant function $d(\mathbf{x}) = \sum_{j=1}^J c_j(\mathbf{x})$, let $t^*$ be such that $\mathrm{P}[\mathcal{S}_{d,t^*}] \geq 1 - q$ under pdf $p(\mathbf{x}) = d(\mathbf{x})/\phi$ . Define $\lambda = (t^*\delta)/(2J(1+\delta))$, and for each*

*j, suppose we have an approximation function $\hat{c}_j$ of $c_j$ such that*

$$|\hat{c}_j(\mathbf{x}) - c_j(\mathbf{x})| \leq \delta/2 c_j(\mathbf{x}), \quad \text{if } c_j(\mathbf{x}) \geq \lambda,$$
$$0 \leq \hat{c}_j(\mathbf{x}) \leq \lambda(1+\delta), \quad \text{otherwise}$$

*Then $\hat{d}(\mathbf{x}) = \sum_j \hat{c}_j(\mathbf{x})$ is a $(\delta, q)$-approximation of $d(\mathbf{x})$ under $p(\cdot)$.*

The proof is presented in Section 3 of the SM. This lemma establishes a sufficient standard of accuracy that we will need for the subnetwork associated with each Gaussian component. In particular, there is a level $\lambda$ such that we need to have relative error better than $\delta/2$ when the component function is greater than $\lambda$. Where the component function is smaller than $\lambda$, we require only an upper bound on the approximation function. The critical level $\lambda$ is proportional to $t^*$, which is a level achieved with high probability by the overall discriminant function $d$. The scaling of the level $t^*$ with $n$ is an important part of the proof, analyzed later in Lemma 5 of the SM.

# 4 Exponential size of SNNs for approximating the GMM discriminant functions

In the previous section, we showed that a DNN with two hidden layers, and $O(n)$ hidden nodes is able to approximate the discriminant functions corresponding to an optimal Bayesian classifier for a collection of GMMs. In this section, we prove a converse result for SNNs. More precisely, we prove that for an SNN to approximate the discriminant function of even a single Gaussian distribution, the number of nodes needs to grow exponentially with $n$.

Consider a neural network with a single hidden layer consisting of $n_1$ nodes. As before, let $\sigma : \mathbb{R} \to \mathbb{R}$ denote the non-linear function applied by each hidden node. For $i = 1, \ldots, n_1$, let $\mathbf{w}_i \in \mathbb{R}^n$, and $b_i \in \mathbb{R}$ denote the weight vector and the bias corresponding to node $i$, respectively. The function generated by this network can be written as

$$f(\mathbf{x}) = \sum_{i=1}^{n_1} a_i \sigma(\langle \mathbf{w}_i, \mathbf{x} \rangle + b_i) + a_0. \tag{6}$$

Suppose that $\mathbf{x} \in \mathbb{R}^n$ is distributed as $\mathcal{N}(\mathbf{0}_n, s_x I_n)$, with pdf $\mu : \mathbb{R}^n \to \mathbb{R}$. Suppose that the function to be approximated is

$$\mu_c(\mathbf{x}) \triangleq \left( \frac{s_f + 2s_x}{s_f} \right)^{\frac{n}{4}} e^{-\frac{1}{2s_f} \|\mathbf{x}\|^2}, \tag{7}$$

which has the form of a symmetric zero-mean Gaussian distribution with variance $s_f$ in each direction, and has been normalized so that $\mathrm{E}[\mu_c^2(\mathbf{x})] = 1$. Our goal is to show that unless the number of nodes $n_1$ is exponentially large in the input dimension $n$, the network cannot approximate the function $\mu_c$ defined in (7) in the sense of Definition 2.

Our result applies to very general activation functions, and allows a different activation function in every node; essentially all we require is that i) the response of each activation function depends on its input $\mathbf{x}$ through a scalar product $\langle \mathbf{w}_i, \mathbf{x} \rangle$, and ii) the output of each hidden node is square-integrable with respect to the Gaussian distribution $\mu(\mathbf{x})$. Incorporating the constant term into one of the activation functions, we consider a more general model

$$f(\mathbf{x}) = \sum_{i=1}^{n_1} a_i h_i(\langle \mathbf{w}_i, \mathbf{x} \rangle) \tag{8}$$

for a set of functions $h_i : \mathbb{R} \to \mathbb{R}$. To avoid scale ambiguities in the definition of the coefficients $a_i$ and the functions $h_i$, we scale $h_i$ as necessary so that, for $i = 1, \ldots, n_1$, $\|\mathbf{w}_i\| = 1$ and $\mathrm{E}[(h_i(\langle \mathbf{w}_i, \mathbf{x} \rangle))^2] = 1$.

Our main result in this section shows the connection between the number of nodes $(n_1)$ and the achievable approximation error $\mathrm{E}[|\mu_c(\mathbf{x}) - f(\mathbf{x})|^2]$. We focus on approximation under Definition 2, which, as shown in Lemma 2, is weaker than the notion used in Theorem 1. Therefore, proving the lower bound under this notion, automatically proves the same bound under the stronger notion too.

**Theorem 2** *Consider $\mu_c : \mathbb{R}^n \to \mathbb{R}$ and $f : \mathbb{R}^n \to \mathbb{R}$ defined in (7) and (8), respectively, for some $s_f > 0$. Suppose that random vector $\mathbf{x} \sim \mathcal{N}(\mathbf{0}_n, s_x I_n)$, where $s_x > 0$. For $i = 1, \ldots, n_1$, assume that $\|\mathbf{w}_i\| = 1$, and $\mathrm{E}[(h_i(\langle \mathbf{w}_i, \mathbf{x} \rangle))^2] = 1$, for activation function $h_i : \mathbb{R} \to \mathbb{R}$. Then,*

$$\mathrm{E}[|\mu_c(\mathbf{x}) - f(\mathbf{x})|^2] \geq 1 - 2\sqrt{n_1} \|\boldsymbol{a}\| \left(1 + s_x/s_f\right)^{1/4} \rho^{-n/4}, \tag{9}$$

*where*

$$\rho \triangleq 1 + \frac{s_x^2}{s_f^2 + 2s_x s_f} > 1. \tag{10}$$

The proof of Theorem 2 is presented in Section 6 of the SM. This result shows that if we want to form an $\epsilon$-relative $\ell_2$ approximation of $\mu_c$, in the sense of Definition 2, with an SNN, $n_1$ must satisfy $n_1 \geq \frac{1-\epsilon}{2A(1+s_x/s_f)^{1/4}} \rho^{n/4}$, where $A = \frac{1}{\sqrt{n_1}} \|\boldsymbol{a}\|$ denotes the root mean-squared value of $\boldsymbol{a}$. That is, the number of nodes need to grow exponentially with $n$, unless the norm of the final layer coefficients vector $\|\boldsymbol{a}\|$ grows exponentially in $n$ as well. Note that in the natural case $s_f = s_x$ where the discriminant function to be approximated matches the distribution of the input data, the required exponential rate of growth is $\rho^{n/4} = (4/3)^{n/4}$.

**Remark 3** *The generalized model* (8) *covers a large class of activation functions. It is straightforward to confirm that the required conditions are satisfied by bounded activation functions, such as the sigmoid function or the* tanh *function, with arbitrary bias values. For the popular ReLU function,* $h_i(\langle \mathbf{w}_i, \mathbf{x} \rangle) = \max(|\langle \mathbf{w}_i, \mathbf{x} \rangle + b_i|, 0)$. *Therefore,* $\mathrm{E}[|h_i(\langle \mathbf{w}_i, \mathbf{x} \rangle)|^2] \leq \mathrm{E}[(\langle \mathbf{w}_i, \mathbf{x} \rangle + b_i)^2] = s_x + b_i^2$, *which again confirms the desired square-integrability property.*

**Remark 4** *From the point of numerical stability, it is natural to require the norm of the final layer coefficients,* $\|\boldsymbol{a}\|$, *to be bounded, as the following simple argument shows. Suppose that network implementation can compute each activation function* $h_i(x)$ *exactly, but that the implementation represents each coefficient* $a_i$ *in a floating point format with a finite precision. To gain intuition on the effect of this quantization noise, consider the following modeling. The implementation replaces* $a_i$ *with* $a_i + z_i$, *where* $\mathrm{E}[z_i] = 0$ *and* $\mathrm{E}[z_i^2] = \nu |a_i|^2$. *Further assume that* $z_1, \ldots, z_{n_1}$ *are independent of each other and of* $\mathbf{x}$. *In this model,* $\nu$ *reflects the level of precision in the representation. Then, the error due to quantization can be written as*

$$\mathrm{E}\left[\sum_i z_i^2 \left(h_i(\langle \mathbf{w}_i, \mathbf{x} \rangle)\right)^2\right] = \sum_i \nu |a_i|^2 \, \mathrm{E}\left[\left(h_i(\langle \mathbf{w}_i, \mathbf{x} \rangle)\right)^2\right] = \nu \|\boldsymbol{a}\|^2 \tag{11}$$

*In such an implementation, in order to keep the quantization error significantly below the targeted overall error* $\epsilon$, *we need to have* $\|\boldsymbol{a}\| \ll \sqrt{\epsilon/\nu}$. *Unless the magnitudes of the weights used in the output layer are bounded in this way, accurate computation is not achievable in practice.*

## 5 Sufficiency of exponentially many nodes

In Section 4, we studied the ability of an SNN in approximating function $\mu_c$ defined as (7) and showed that such a network, if the weights are not allowed to grow exponentially with $n$, requires exponentially many nodes to make the error small. Clearly, Theorem 2 is a converse result, which implies that the number of nodes $n_1$ should grow with $n$, at least as $\rho^{\frac{n}{4}}$ ($\rho > 1$). The next natural question is the following: Would exponentially many nodes actually suffice in order to approximate function $\mu_c$? In this section, we answer this question affirmatively and show a simple construction with random weights that, given enough nodes, is able to well approximate function $\mu_c$ defined in (7), within the desired accuracy. Recall that $\mu_c(\mathbf{x}) = \alpha^{\frac{n}{4}} \exp(-\frac{1}{2s_f} \|\mathbf{x}\|^2)$, where $\alpha \triangleq \frac{s_f + 2s_x}{s_f}$. Consider the output function of a single-hidden layer neural network with all biases set to zero. The function generated by such a network can be written as

$$f(\mathbf{x}) = \sum_{i=1}^{n_1} a_i \sigma(\langle \mathbf{w}_i, \mathbf{x} \rangle). \tag{12}$$

As before, here, $\sigma : \mathbb{R} \to \mathbb{R}$ denotes the non-linear function and $\mathbf{w}_i \in \mathbb{R}^n$, $\|\mathbf{w}_i\| = 1$, denotes the weights used by hidden node $i$. To show sufficiency of exponentially many nodes, we consider a special non-linear function $\sigma(x) = \cos(x/\sqrt{s_f})$.

**Theorem 3** *Consider function* $\mu_c : \mathbb{R}^n \to \mathbb{R}$, *defined in* (7), *and n-dimensional random vector* $\mathbf{x} \sim \mathcal{N}(\mathbf{0}_n, s_x I_n)$. *Consider function* $f : \mathbb{R}^n \to \mathbb{R}$ *defined in* (12). *Let* $\sigma(x) = \cos(x/\sqrt{s_f})$, *and, for* $i = 1, \ldots, n_1$, $a_i = \alpha^{\frac{n}{4}}/n_1$, *where* $\alpha = 1 + 2s_x/s_f$. *Given* $\epsilon > 0$, *assume that* $n_1 > \frac{1}{\epsilon} \alpha^{\frac{n}{2}}$. *Then, there exists weights* $\mathbf{w}_1, \ldots, \mathbf{w}_{n_1}$ *such that* $\mathrm{E}_{\mathbf{x}}[(f(\mathbf{x}) - \mu_c(\mathbf{x}))^2] \leq \epsilon$.

The Proof of Theorem 3 is presented in Section 7 of the SM. To better understand the implications of Theorem 3 and how it compares against Theorem 2, define $m_1 = \rho = 1 + \frac{s_x}{2s_f}(1 - \frac{s_f}{s_f+2s_x})$ and $m_2 = \alpha^2 = 1 + \frac{4s_x}{s_f}(1 + \frac{s_x}{s_f})$, where $\rho$ is defined in (10). It is straightforward to see that $1 < m_1 < m_2$, for all positive values of $(s_x, s_f)$. Theorems 2 and 3 show that there exist constants $c_1$ and $c_2$, such that if the number of hidden nodes in a single-hidden-layer network $(n_1)$ is smaller than $c_1 m_1^{n/4}$, the expected error in approximating function $\mu_c(\mathbf{x})$ must get arbitrarily close to one. On other hand, if $n_1$ is larger than $c_2 m_2^{n/4}$, then there exists a set of weights such that the error can be made arbitrary close to zero. In other words, it seems that there is a phase transition in the exponential growth of the number of nodes, below which, the function cannot be approximated with a single hidden layer. Characterizing that phase transition is an interesting open question, which we leave to future work.

# 6  Related work

There is a rich and well-developed literature on the complexity of Boolean circuits, and the important role depth plays in them. However, since it is not clear to what extend such results on Boolean circuits has a consequence for DNNs, we do not summarize this literature. The interested reader may wish to start with [16]. A key notion for us is that of depth, that is to so say, the number of (hidden) layers of nodes in a neural network as defined in Section 1.2. We are interested to know to what extent, if any, depth reduces complexity of the neural network to express or approximate functions of interest in classification. It is not the complexity of the function that we want to approximate that matters, because the UAT already tells us that regular functions, which include discriminant functions we discuss in Section 1.3, can be approximated by SNNs, shallow neural networks. But the complexity of the NNs, as measured by the *number* of nodes needed for the approximation is of interest to us. In this respect, the work of [17, 18, 19] contain approximation results for neural structures for certain polynomials and tensor functions, in the spirit of what we are looking for, but as with Boolean circuits, these models deviate substantially from the standard DNN models we consider here, those that represent the neural networks that have worked well in practice and for whose behavior we wish to obtain fundamental insights.

Remarkably, there is a small collection of recent results which, as in this paper, show that adding a single layer to an SNN reduces the number of nodes by a logarithmic factor for approximation of some special functions: see [13, 11, 12, 20] for approximation of high-degree polynomials, a certain radial function, special functions of the inner products of high-dimensional vectors, and saw-tooth functions, respectively. Our work is therefore in the same spirit as these, showing *the power of two* in the reduction of complexity of DNNs, and is therefore, the continuation and generalization of the said set of results and is especially informed by [11]. For a specialized radial function in $R^n$, [11] shows that while any SNN would require at least exponentially many nodes to approximate the function, there exists a DNN with two hidden layers and $O(n^{19/4})$ nodes that well approximates the same function. In the present work, for a general class of widely-used functions viz GMM discriminant functions, we show that while SNNs require at least exponentially many nodes, for any GMM discriminant function there exists a DNN with two hidden layers and only $O(n)$ nodes that approximates it.

# 7  Remarks and conclusion

It is worth noting that even though to prove Theorem 1 we used a variety of sufficient regularity assumptions for the non-linear function $\sigma$, these assumptions are not necessary to construct an efficient two-layer network. For example, to construct a network using the commonly used Rectifier Linear Unit (ReLu) activation, in the first layer we can form $n$ super-nodes, each of which has a piecewise constant response $h_i(x)$ that approximates $x^2$ with the accuracy specified in Lemma 1 of the SM. The number of basic nodes needed in each super-node in this construction is $2R/\sqrt{\nu}$, where $R$ and $\nu$ denote the range and the accuracy for approximating $x^2$ in layer one, respectively. The analysis of $R$ and $\nu$ in Lemmas 3 and 5 in the SM show that $R$ is $O(\sqrt{n})$ and $\nu$ is $O(1/n)$, so that the number of nodes needed per super-node in the first layer is now $O(n)$, compared to $O(1)$ in the construction presented in Section 3. Since there are $n$ such nodes, the total number of basic nodes in the network becomes $O(n^2)$ - still an exponential reduction compared with a single layer network.

## Footnotes

[1]Throughout the paper, a Bayesian classifier refers to a classifier that has access to the distribution of the data.

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
