[Supplementary Material · DL_NIPS2019_v3_supp.pdf]

# Efficient Deep Approximation of GMMs: Supplementary material

**Shirin Jalali, Carl Nuzman, Iraj Saniee**
Bell Labs, Nokia
600-700 Mountain Avenue
Murray Hill, NJ 07974
{shirin.jalali,carl.nuzman,iraj.saniee}@nokia-bell-labs.com

## 1   Proof of Lemma 1

Note that

$$
\begin{aligned}
e_{21} &= \mathrm{P}_1(\hat{p}_2(\mathbf{x}) \geq \hat{p}_1(\mathbf{x})) = \mathrm{P}_1(\hat{p}_2(\mathbf{x}) \geq \hat{p}_1(\mathbf{x}), \mathbf{x} \in \mathcal{S}_{p_1,t_1}) + \mathrm{P}_1(\hat{p}_2(\mathbf{x}) \geq \hat{p}_1(\mathbf{x})), \mathbf{x} \in \mathcal{S}^c_{p_1,t_1}) \\
&\leq \mathrm{P}_1(\mathcal{S}^c_{p_1,t_1}) + \mathrm{P}_1(\hat{p}_2(\mathbf{x}) \geq \hat{p}_1(\mathbf{x}), \mathbf{x} \in \mathcal{S}_{p_1,t_1}) \\
&\leq q + \mathrm{P}_1(\hat{p}_2(\mathbf{x}) \geq \hat{p}_1(\mathbf{x}), \mathbf{x} \in \mathcal{S}_{p_1,t_1}, \mathbf{x} \in \mathcal{S}_{p_2,t_2}) + \mathrm{P}_1(\hat{p}_2(\mathbf{x}) \geq \hat{p}_1(\mathbf{x}), \mathbf{x} \in \mathcal{S}_{p_1,t_1}, \mathbf{x} \in \mathcal{S}^c_{p_2,t_2}) \\
&\leq q + \mathrm{P}_1((1+\delta)p_2(\mathbf{x}) \geq (1-\delta)p_1(\mathbf{x})) + \mathrm{P}_1(\hat{p}_2(\mathbf{x}) \geq \hat{p}_1(\mathbf{x}), \mathbf{x} \in \mathcal{S}_{p_1,t_1}, \mathbf{x} \in \mathcal{S}^c_{p_2,t_2}) \\
&\leq q + \mathrm{P}_1(\frac{1+\delta}{1-\delta}p_2(\mathbf{x}) \geq p_1(\mathbf{x})) + \mathrm{P}_1((1+\delta)t_2 \geq (1-\delta)p_1(\mathbf{x})),
\end{aligned}
\tag{1}
$$

which yields the desired result.

## 2   Proof of Lemma 2

Let $t$ be the threshold in the definition of $(\delta, q)$ approximation. Where $p$ exceeds $t$, we have $\mathrm{E}[(\hat{p}-p)^2 \,|\, \mathcal{S}_{p,t}] \leq \delta^2 \,\mathrm{E}[p^2 \,|\, \mathcal{S}_{p,t}]$, and where $p$ is less than $t$, we have $\mathrm{E}[(\hat{p}-p)^2 \,|\, \mathcal{S}^c_{p,t}] \leq (1+\delta)^2 t^2$. We also know that

$$
\mathrm{E}[p^2] \geq \mathrm{E}[p^2 \,|\, \mathcal{S}_{p,t}]\,\mathrm{P}[\mathcal{S}_{p,t}] \geq t^2(1-q)
$$

Hence, we have

$$
\begin{aligned}
\mathrm{E}[(\hat{p}-p)^2] &= \mathrm{E}[(\hat{p}-p)^2 \,|\, \mathcal{S}_{p,t}]\,\mathrm{P}[\mathcal{S}_{p,t}] + \mathrm{E}[(\hat{p}-p)^2 \,|\, \mathcal{S}^c_{p,t}]\,\mathrm{P}[\mathcal{S}^c_{p,t}] \\
&\leq \delta^2 \,\mathrm{E}[p^2 \,|\, \mathcal{S}_{p,t}]\,\mathrm{P}[\mathcal{S}_{p,t}] + (1+\delta)^2 t^2 q \\
&\leq \delta^2 \,\mathrm{E}[p^2] + \frac{(1+\delta)^2 q}{1-q}\,\mathrm{E}[p^2].
\end{aligned}
$$

## 3   Proof of Lemma 3

First, suppose that $x \in \mathcal{S}_{d,t^*}$. We have

$$
\begin{aligned}
|\hat{d}(\mathbf{x}) - d(\mathbf{x})| &\leq \sum_j |\hat{c}_j(\mathbf{x}) - c_j(\mathbf{x})| \\
&\leq \sum_{j:c_j(\mathbf{x}) \geq \lambda} |\hat{c}_j(\mathbf{x}) - c_j(\mathbf{x})| + \sum_{j:c_j(\mathbf{x}) < \lambda} |\hat{c}_j(\mathbf{x}) - c_j(\mathbf{x})|
\end{aligned}
$$

Consider the first partial sum, over $j$ for which $c_j(\mathbf{x}) \geq \lambda$. By assumption each term in the sum is bounded by $\delta/2 c_j(\mathbf{x})$, and so the partial sum is upper bounded by $\delta/2d(\mathbf{x})$. Considering the

second partial sum, since $0 \leq c_j(\mathbf{x}) < \lambda$ and $0 \leq \hat{c}_j(\mathbf{x}) \leq \lambda(1 + \delta)$, each term is upper bounded by $\lambda(1 + \delta)$, and the partial sum upper-bounded by $J\lambda(1 + \delta) = t^*\delta/2$. Since $d(\mathbf{x}) \geq t^*$, the sum is upper-bounded by $\delta/2d(\mathbf{x})$. Putting both sums together, $|\hat{d}(\mathbf{x}) - d(\mathbf{x})| \leq \delta d(\mathbf{x})$. Thus $\hat{d}$ has the required relative accuracy on $\mathcal{S}_{d,t^*}$.

Now, suppose $\mathbf{x} \notin \mathcal{S}_{d,t^*}$, i.e. $t^* > d(\mathbf{x}) \geq c_j(\mathbf{x})$. If $c_j(\mathbf{x}) < \lambda$, then $\hat{c}_j(\mathbf{x}) \leq \lambda(1 + \delta) \leq t^*\delta/(2J)$. If $c_j(\mathbf{x}) \geq \lambda$, then $\hat{c}_j(\mathbf{x}) \leq (1 + \delta/2)c_j(\mathbf{x})$. Putting both together, we have

$$\hat{c}_j(\mathbf{x}) \leq (1 + \delta/2)c_j(\mathbf{x}) + \frac{t^*\delta}{2J}$$

and

$$
\begin{aligned}
\sum_j \hat{c}_j(\mathbf{x}) &\leq (1 + \delta/2)d(\mathbf{x}) + \frac{t^*\delta}{2} \\
&\leq (1 + \delta)t^*.
\end{aligned}
$$

## 4   Lemmas supporting Theorem 1

In this section, we state and prove some technical lemmas used in the proof of Theorem 1.

**Lemma 1** *Given range $R > 0$ and desire accuracy $\nu > 0$, let $h(x, a)$, defined in (2), be constructed from an activation function $\sigma$ satisfying Assumptions 1 and 2, with $r$, $M$, and $\tau$ as defined in those assumptions. Choose $a$ satisfying $a \geq 2R/r$, $a \geq 8MR^3/(3\nu\sigma^{(2)}(\tau))$ and $a \geq 4MR/(3\sigma^{(2)}(\tau))$. Then*

- *$h(x, a) \geq 0$ for all $x$, and*
- *$|h(x, a) - x^2| < \nu$ for $|x| \leq 2R$, and*
- *$h(x, a) \geq 4R^2 - \nu$ for $|x| > 2R$.*

**Proof:** From Assumption 1, for $|x - \tau| \leq r$, $\sigma$ has three derivatives and in this neighborhood, there is an $M' = 2M/\sigma^{(2)}(\tau)$ such that $|\sigma^{(3)}(x)| \leq M = M'\sigma^{(2)}(\tau)/2$. Then in the interval $|x| < ar$, $h(x, a)$ has three derivatives and the third derivative is bounded by $M'/a$. Moreover, $h(0, a) = h'(0, a) = 0$, and the 2nd-order Taylor polynomial of $h(x, a)$ at zero is simply $T_2(x) = x^2$. From Taylor's theorem, it follows that

$$|h(x, a) - x^2| \leq \frac{M'}{3!a}|x|^3$$

on $|x| \leq ar$.

Now, $|x| \leq 2R$ implies $|x| \leq ar$. Using Taylor's theorem and the second constraint on $a$, we have

$$|h(x, a) - x^2)| \leq \frac{M' \cdot 3\nu}{6 \cdot 4M'R^3}|x|^3 \leq \nu,$$

as desired. Also, on this interval, using Taylor's theorem and the third constraint on $a$, we have

$$h(x, a) \geq |x|^2 - \frac{M' \cdot 3}{6 \cdot 2M'R}|x|^3 = |x|^2(1 - |x|/(4R)) \geq (1/2)|x|^2 \geq 0,$$

establishing the non-negativity of $h$ on this interval.

To show $h(x, a) \geq 4R^2 - \nu$, for $|x| > 2R$, we rely on Assumption 2, which implies that $h(x, a)$ is monotonically increasing for $x \geq 0$. For $|x| > 2R$, we have $h(x, a) = h(|x|, a) \geq h(2R, a) \geq 4R^2 - \nu$. $\square$

Summing the outputs of $n$ supernodes, we obtain $\hat{g}(\mathbf{x}) = \sum_{i=1}^n h(x_i, a)$, which is an approximation to $g(\mathbf{x})$, as expressed in the following corollary.

**Corollary 1** *Given a function $h(x, a)$ and associated range $R > 0$ and accuracy $\nu > 0$ as defined in Lemma 1, the function $\hat{g}(\mathbf{x}) = \sum_{i=1}^n h(x_i, a)$ satisfies*

- $\hat{g}(\mathbf{x}) \geq 0$

- $|\hat{g}(\mathbf{x}) - g(\mathbf{x})| \leq n\nu$ when $g(\mathbf{x}) \leq 4R^2$

- $\hat{g}(\mathbf{x}) \geq 4R^2 - n\nu$ when $g(\mathbf{x}) > 4R^2$

**Proof:** The first statement is trivial. To see the second statement, suppose $g(\mathbf{x}) \leq 4R^2$. Then $|x_i| \leq 2R$ for each $i$ and $|h(x_i, a) - x_i^2| \leq \nu$ for each $i$, yielding the result. For the second statement, suppose $g(\mathbf{x}) > 4R^2$. If $|x_i| > 2R$ for some $i$, then $h(x_i, a) \geq 4R^2 - \nu$, and non-negativity of $h$ implies $\hat{g}(\mathbf{x}) \geq 4R^2 - \nu \geq 4R^2 - n\nu$. On the other hand, if $|x_i| < 2R$ for all $i$, then $h(x_i, a) \geq x_i^2 - \nu$ for all $i$, and so $\hat{g}(\mathbf{x}) \geq g(\mathbf{x}) - n\nu$. $\square$

**Lemma 2** *Given an activation function $\sigma$, defined in* (4)*, satisfying Assumption 3 with $\eta = 1$, a range $T > 0$ and accuracy $\epsilon > 0$, construct the supernode function $\psi(x, \sigma)$ using $K$ basic nodes, where $K$ is chosen so that $\Delta = T/K$ satisfies $\Delta \leq \log(1 + \epsilon/40)$ and $\Delta < 1/2$. Then this function satisfies*

- $|\psi(x, \sigma) - \exp(-x)| \leq \epsilon \exp(-x)$ *for* $0 \leq x \leq T$

- $0 \leq \psi(x, \sigma) \leq \exp(-T)(1 + \epsilon)$ *for* $x \geq T$

**Proof:** Note that if $\sigma(x)$ is an indicator function $u(x) := 1\{x \geq 0\}$, this construction simply gives a piecewise-constant step approximation to $\exp(-x)$ over the interval $[0, T]$. The relative accuracy of such an approximation is uniform over the interval when using steps of fixed width $\Delta$. In particular, in the interval $k\Delta \leq x \leq (k+1)\Delta$, with $k \leq K$, the function $\psi(x, u)$ is equal to $\exp(-k\Delta)$, and the worst-case relative error for $x \in [0, T]$ is $(\exp(-k\Delta) - \exp(-(k+1)\Delta)/\exp(-(k+1)\Delta) = \exp(\Delta) - 1$. Then $\Delta \leq \log(1 + \epsilon/40)$, ensures that the relative error between $\psi(x, u)$ and $\exp(-x)$ is no more than $\epsilon/40$ on $[0, T]$.

For a general activation function we can write
$$|\psi(x, \sigma) - \exp(-x)| \leq |\psi(x, u) - \exp(-x)| + |\psi(x, \sigma) - \psi(x, u)|.$$
The relative accuracy of $\psi(x, \sigma)$ is thus assured if we can show e.g. that $|\psi(x, \sigma) - \psi(x, u)| \leq \epsilon/2 \exp(-x)$ on $[0, T]$. By virtue of Assumption 3, we know that $|\sigma(x) - u(x)| \leq \exp(-|x|)$. Hence we can write
$$|\psi(x, \sigma) - \psi(x, u)| \leq \left(e^\Delta - 1\right) \sum_{k=1}^{K} e^{-|x/\Delta - k| - k\Delta}.$$

The sum can be further upperbounded by summing over all $-\infty < k < \infty$. Denoting $m = \lfloor x/\Delta \rfloor$ and $\nu = x/\Delta - m$, we can break the sum into ranges $k \leq m$ and $k > m$. The first range gives
$$\sum_{k \leq m} e^{-m-\nu+(1-\Delta)k} \quad \leq \quad e^{-\nu - \Delta m} \sum_{j \geq 0} e^{-(1-\Delta)j}$$
$$\leq \quad \frac{e^{-x+\Delta}}{1 - e^{-(1-\Delta)}} = e^{-x} \frac{e^\Delta}{1 - e^{-(1-\Delta)}}.$$
Likewise, the second range gives
$$\sum_{k > m} e^{m+\nu-(1+\Delta)k} \quad \leq \quad e^{-(1+\Delta)+\nu - \Delta m} \sum_{j \geq 0} e^{-(1+\Delta)j}$$
$$\leq \quad e^{-x} \frac{e^1}{1 - e^{-(1+\Delta)}} \leq e^{-x} \frac{e^1}{1 - e^{-1}}$$
Using $\Delta < 1/2$, we can combine terms to get
$$|\psi(x, \sigma) - \psi(x, u)| \leq \left(e^\Delta - 1\right) \frac{2e^1}{1 - e^{-1/2}} e^{-x}$$

Since $\Delta \leq \log(1 + (\epsilon/40)) \leq \log(1 + (\epsilon/2)(1 - e^{-1/2})/(2e))$, we obtain $|\psi(x, \sigma) - \psi(x, u)| \leq \epsilon/2 e^{-x}$, which completes the proof of relative accuracy on $[0, T]$.

To show that $\psi(x, \sigma) \leq \exp(-T)(1 + \epsilon)$ for $x \geq T$, we note that $\psi(x, u) = exp(-T)$ for $x \geq T$. We have already shown that $|\psi(x, \sigma) - \psi(x, u)| \leq \epsilon/2 e^{-x}$ for all $x$, and hence $|\psi(x, \sigma) - \psi(x, u)| \leq \epsilon/2 e^{-T}$ for $x \geq T$ in particular. $\square$

**Lemma 3** *Given desired accuracy $0 < \delta < 2$ and level $0 < \lambda < 1/(1 + \frac{\delta}{4})$,*

- *Let the function $\hat{g}(\mathbf{x}) = \sum_{i=1}^{n} h(x_i, a)$ be the function defined in Corollary 1 with range parameter $R = \sqrt{\log(1/\lambda)}$ and accuracy parameter $\nu = 1/n \log(1 + \delta/4)$.*

- *Let $\psi(x, \sigma)$ be a function satisfying conditions of Lemma 2 with range parameter $T = 4R^2$ and accuracy $\epsilon = \delta/2$.*

- *Define $\hat{c}(\mathbf{x}) = \psi(\hat{g}(\mathbf{x}), \sigma)$.*

*Then*

- $|\hat{c}(\mathbf{x}) - c(\mathbf{x})| < \delta c(\mathbf{x})$ *whenever $c(\mathbf{x}) \geq \lambda$*

- $\hat{c}(\mathbf{x}) < \lambda(1 + \delta)$ *whenever $c(\mathbf{x}) < \lambda$.*

**Proof:** First, let us suppose that $c(\mathbf{x}) \geq \lambda$. This implies $g(\mathbf{x}) \leq R^2 < 4R^2$. Hence, by Corollary 1, $|\hat{g}(\mathbf{x}) - g(\mathbf{x})| \leq n\nu = \log(1 + \delta/4)$, and so $|\exp(-\hat{g}(\mathbf{x})) - c(\mathbf{x})| \leq \delta/4\, c(\mathbf{x})$. Moreover, since $\hat{g}(\mathbf{x}) \leq g(\mathbf{x}) + \log(1 + \delta/4)$, $g(\mathbf{x}) \leq R^2$, and $\log(1 + \delta/4) \leq \log \frac{1}{\lambda} \leq R^2$, we have $\hat{g}(\mathbf{x}) \leq 2R^2 \leq T$. Thus by Lemma 2, $|\hat{c}(\mathbf{x}) - \exp(-\hat{g}(\mathbf{x}))| = |\psi(\hat{g}(\mathbf{x}), \sigma) - \exp(-\hat{g}(\mathbf{x}))| \leq \delta/2 \exp(-\hat{g}(\mathbf{x}))$. Thus

$$
\begin{aligned}
|\hat{c}(\mathbf{x}) - c(\mathbf{x})| &\leq |\hat{c}(\mathbf{x}) - \exp(-\hat{g}(\mathbf{x}))| + |\exp(-\hat{g}(\mathbf{x})) - c(\mathbf{x})| \\
&\leq \frac{\delta}{4} c(\mathbf{x}) + \frac{\delta}{2} \exp(-\hat{g}(\mathbf{x})) \\
&\leq \frac{\delta}{4} c(\mathbf{x}) + \frac{\delta}{2}(1 + \frac{\delta}{4}) c(\mathbf{x}) \\
&\leq \delta\, c(\mathbf{x}),
\end{aligned}
$$

where in the last step we use $\delta < 2$.

Secondly, suppose that $c(\mathbf{x}) < \lambda$, so that $g(\mathbf{x}) > R^2$. Whether or not $g(\mathbf{x}) \geq 4R^2$, Corollary 1 implies that $\hat{g}(\mathbf{x}) \geq R^2 - n\nu = -\log(\lambda(1 + \delta/4))$, so that $\exp(-\hat{g}(\mathbf{x})) \leq \lambda(1 + \delta/4)$. If $\hat{g}(\mathbf{x}) \leq T$, then (second layer result, Lemma 2) gives $\psi(\hat{g}(\mathbf{x})) \leq \epsilon \exp(-\hat{g}(\mathbf{x})) \leq \lambda(1 + \delta/4)\delta/2 \leq \lambda(1 + \delta)$ assuming $\delta < 4$. Or, if $\hat{x}(\mathbf{x}) > T$, it follows from Lemma 2 that $\psi(\hat{g}(\mathbf{x})) \leq \exp(-T)(1 + \delta/2) \leq \exp(-R^2)(1 + \delta/2) \leq \lambda(1 + \delta)$ as desired. $\square$

**Lemma 4** *Let $\mathbf{x}$ be a multi-dimensional Gaussian random variable in $\mathbb{R}^n$ with pdf $p(\mathbf{x})$, and suppose that $\mathrm{var}(X_i) \leq V$ for each component. Then*

$$
P\left[p(\mathbf{x}) < t\right] \leq t^4 \exp\left(\frac{n}{8} \log(32\pi V)\right)
$$

**Proof:** The pdf can be written as $p(\mathbf{x}) = ((2\pi)^n |\Sigma|)^{-1/2} \exp(-g(\mathbf{x}))$ where $g(\mathbf{x}) = (\mathbf{x} - \boldsymbol{\mu})^T \Sigma^{-1} (\mathbf{x} - \boldsymbol{\mu})$. Hence

$$
\mathrm{P}[p(\mathbf{x}) < t] = \mathrm{P}\left[g(\mathbf{x}) > \log(1/t) - \frac{1}{2}\log|\Sigma| - \frac{n}{2}\log(2\pi)\right].
$$

The eigenvalues of $\Sigma$ are positive and bounded as $\sum_{i=1}^{n} \lambda_i = \mathrm{trace}(\Sigma) \leq nV$. Maximizing $|\Sigma| = \Pi_i \lambda_i$ under this constraint, via Lagrange multipliers, yields $|\Sigma| \leq V^n$. Thus

$$
\mathrm{P}[p(\mathbf{x}) < t] \leq \mathrm{P}[g(\mathbf{x}) > \log(1/t) - \frac{n}{2}\log(2\pi V)].
$$

For $\mathbf{x} \sim \mathcal{N}(\boldsymbol{\mu}, \Sigma)$, $g(\mathbf{x})$ is a standard chi-squared variate with $n$ degrees of freedom. The Chernoff bound for such a random variable can be expressed as

$$
\mathrm{P}\left[g(\mathbf{x}) \geq (1 + \theta)n\right] \leq \exp\left(-\frac{n}{2}(\theta - \log(1 + \theta))\right).
$$

Using a tangent bound to a convex function at $\theta = 1$, we have $\theta - \log(1 + \theta) \geq (\theta + 1)/2 - \log(2)$, so that

$$
\mathrm{P}\left[g(\mathbf{x}) \geq (1 + \theta)n\right] \leq \exp\left(-\frac{n}{4}(\theta + 1 - 2\log(2))\right).
$$

and

$$P\left[g(\mathbf{x}) \geq s\right] \leq \exp\left(-\frac{1}{4}(s - 2n\log(2))\right).$$

Putting the bounds together, yields

$$P\left[p(\mathbf{x}) < t\right] \leq \exp\left(-\frac{1}{4}\left(\log(1/t) - \frac{n}{2}\log(2\pi V) - \frac{n}{2}\log(16)\right)\right)$$

$$\leq t^4 \exp\left(\frac{n}{8}\log(32\pi V)\right)$$

as desired. □

We can now extend the analysis to a GMM.

**Lemma 5** *Let $p(\mathbf{x}) = \sum_{j=1}^{J} \alpha_j p_j(\mathbf{x})$ be the pdf of an $n$-dimensional GMM, where $\alpha_j$ and $p_j(\mathbf{x})$, $j = 1, \ldots, J$, denote the probability and the pdf of the Gaussian distribution $j$, respectively. Define $\mathbf{x}$ to be a random vector with distribution $p$. Assume that $\operatorname{var}(x_i) \leq 1$, $i = 1, \ldots, n$, for each element of $\mathbf{x}$. Given $q > 0$, choose $t^* > 0$ such that*

$$\log(1/t^*) \geq \frac{n}{32}\log(32\pi J) + \frac{1}{4}\log(1/q) + \log(J).$$

*Then*

$$P[p(\mathbf{x}) < t^*] \leq q.$$

**Proof:** Choose $k = \arg\max_j \alpha_j$; we have $\alpha_k \geq 1/J$. Since $\operatorname{var}(x_i) \geq \sum_j \alpha_j \operatorname{var}(x_i^{(j)})$, where $\mathbf{x}^{(j)}$ denotes the j-th Gaussian variable in the mixture, we have $\operatorname{var}(x_i^{(k)}) \leq 1/\alpha_k \leq J$. Now applying Lemma 4,

$$\begin{aligned} P\left[p(\mathbf{x}) < t^*\right] &\leq P\left[\alpha_k p_k(\mathbf{x}) < t^*\right] \\ &\leq P\left[p_k(\mathbf{x}) < J t^*\right] \\ &\leq (J t^*)^4 \exp\left(\frac{n}{8}\log(32\pi J)\right) \leq q, \end{aligned}$$

where the last line follows from the assumed upper bound on $\log(1/t^*)$. □

## 5   Proof of Theorem 1

We provide an explicit construction of a two-hidden layer subnetwork for approximating a given $c_j(\mathbf{x})$, and show that under Assumptions 1, 2, and 3, the constructed network is accurate enough to satisfy the conditions of Lemma 3 in the main paper. This is done by showing that super-nodes in the first layer approximate $x^2$ well enough, and a super-node in the second layer approximates $\exp(-x)$ well enough.

In the rest of the proof we drop the subscript $j$ and focus on a $c(\mathbf{x}) = \beta\exp(-g(\mathbf{x}))$, where $g(\mathbf{x}) = \sum_i y_i^2$ with $\mathbf{y} = \Sigma^{-1/2}(\mathbf{x} - \boldsymbol{\mu})$. Given parameters $\delta > 0$ and $\lambda > 0$, we construct an approximation $\hat{c}$ of $c$, such that i) $|\hat{c}(\mathbf{x}) - c(\mathbf{x})| < \delta c(\mathbf{x})$, whenever $c(\mathbf{x}) \geq \lambda$, and ii) $|\hat{c}(\mathbf{x}) - c(\mathbf{x})| < \lambda(1 + \delta)$, whenever $c(\mathbf{x}) < \lambda$. We show that the total number of nodes used by both hidden layers is $O(n)$.

To simplify the following steps, we normalize by $\beta$ to obtain $\tilde{c}(\mathbf{x}) = c(\mathbf{x})/\beta$ which is to be approximated accurately above level $\tilde{\lambda} = \lambda/\beta$. If $\tilde{\lambda} \geq 1$, then it is sufficient to simply take the trivial approximation $\hat{\tilde{c}}(\mathbf{x}) = 0$, since $\tilde{c}(x) \leq 1$ everywhere. Therefore in the following we assume $\tilde{\lambda} < 1$. With notation thus simplified, we seek to approximate $\tilde{c}(\mathbf{x}) = \exp(-\sum_i y_i^2)$.

The condition $\tilde{c}(\mathbf{x}) \geq \tilde{\lambda}$ corresponds to $\sum_i y_i^2 \leq \log(1/\tilde{\lambda})$. We will define $R = \sqrt{\log(1/\tilde{\lambda})}$, so that $\sum_i y_i^2 \leq R^2$ defines the region over which we must approximate $\tilde{c}(\mathbf{x})$ with small relative error.

*Uniform approximation of $x^2$.*   In the first hidden layer, we replace each node with activation function $x^2$ in the ideal reference model, with a supernode formed from two basic nodes with

activation function $\sigma$ satisfying Assumptions 1 and 2. We use a special case of the construction in [1]. In particular, we define the supernode as

$$h(x, a) = \frac{a^2}{\sigma^{(2)}(\tau)} \left( \sigma(x/a + \tau) + \sigma(-x/a + \tau) - 2\sigma(\tau) \right). \tag{2}$$

The idea behind the construction is that the first term in the Taylor series of $h(x, a)$ at $x = 0$ is $x^2$, and so for sufficiently large $a$, the function approximates $x^2$ closely over the required range. In particular, by Lemma 1, choosing $a \geq \max(2R/r, 8MR^3/(3\nu\sigma^{(2)}(\tau)), 4MR/(3\sigma^{(2)}(\tau)))$, we have

- $h(x, a) \geq 0$ for all $x$, and
- $|h(x, a) - x^2| < \nu$ for $|x| \leq 2R$, and
- $h(x, a) \geq 4R^2 - \nu$ for $|x| > 2R$.

Summing the outputs of $n$ supernodes, we obtain

$$\hat{g}(\mathbf{x}) = \sum_{i=1}^{n} h(y_i, a). \tag{3}$$

Then, by Corollary 1, $\hat{g}(\mathbf{x})$ is an approximation to $g(\mathbf{x})$, such that

- $|\hat{g}(\mathbf{x}) - g(\mathbf{x})| \leq n\nu$, when $g(\mathbf{x}) \leq 4R^2$,
- $\hat{g}(\mathbf{x}) \geq 4R^2 - n\nu$, when $g(\mathbf{x}) > 4R^2$.

*Approximation of* $\exp(-x)$. After approximating the quadratic term $g(\mathbf{x})$ in the first layer, the role of the next level is to approximate $\exp(-x)$, with a required level of accuracy $\epsilon$, over a required range $[0, T]$, using only $O(n)$ nodes.

For this construction, we rely on Assumption 3. The value of the bounding exponent $\eta$ in this assumption is not critical, since if $\sigma(x)$ satisfies the assumption with exponent $\eta$, the scaled function $\sigma(\alpha x)$ satisfies it with exponent $\alpha\eta$. To simplify notation, we take $\eta = 1$.

To form the super-node, we choose a sufficiently large number of components $K$, divide the range into intervals of width $\Delta = T/K$, and then form following sum of shifted activations:

$$\psi(x, \sigma) = 1 + \sum_{k=1}^{K} \left( e^{-k\Delta} - e^{-(k-1)\Delta} \right) \sigma\left( \frac{x}{\Delta} - k \right). \tag{4}$$

This essentially constructs a staircase-like approximation to $\exp(-x)$. Lemma 2 states that if $\Delta \leq \min(\log(1 + \epsilon/40), 1/2)$, then

- $|\psi(x, \sigma) - \exp(-x)| \leq \epsilon \exp(-x)$, for $0 \leq x \leq T$,
- $0 \leq \psi(x, \sigma) \leq \exp(-T)(1 + \epsilon)$, for $x \geq T$.

*Accuracy of composed layers.* Consider $\hat{g}$ defined in (3), with range parameter $R = \sqrt{\log(1/\tilde{\lambda})}$ and accuracy parameter $\nu = \frac{1}{n}\log(1 + \delta/4)$. Also, consider $\psi(x, \sigma)$ defined in (4), with range parameter $T = 4R^2$ and accuracy $\epsilon = \delta/2$. Define

$$\hat{\tilde{c}}(\mathbf{x}) = \psi(\hat{g}(\mathbf{x}), \sigma). \tag{5}$$

By Lemma 3,

- $|\hat{\tilde{c}}(\mathbf{x}) - \tilde{c}(\mathbf{x})| < \delta\tilde{c}(\mathbf{x})$, whenever $\tilde{c}(\mathbf{x}) \geq \tilde{\lambda}$,
- $\hat{\tilde{c}}(\mathbf{x}) < \tilde{\lambda}(1 + \delta)$, whenever $\tilde{c}(\mathbf{x}) < \tilde{\lambda}$.

We have thus established that, for arbitrarily small $\tilde{\lambda} > 0$, we can construct an approximation $\hat{\tilde{c}}(\mathbf{x})$ with the accuracy required by Lemma 3 in the main paper. The same statement evidently holds for $\hat{c}(\mathbf{x}) = \beta\hat{\tilde{c}}(\mathbf{x})$ with respect to $\lambda = \beta\tilde{\lambda}$. Hence by Lemma 3 in the main body, we can construct a

$(\delta, q)$-approximation $\hat{d}$ of $d$, for any $\delta$ and $q$. It remains to show that that the number of nodes required, $M_n$, is $O(n)$ and that the coefficient sizes grow no faster than $O(n^5)$.

*Bounding the number of hidden nodes.* The final step is to show that the described network giving $\hat{c}(\mathbf{x})$ consists of $O(n)$ nodes. Exactly $2n$ nodes are needed in the first layer, since each of the $n$ functions $h_i(x, a)$ is constructed with two nodes. In the second layer, the number of nodes is $K = T/\Delta$.

Since our network is designed with $\epsilon = \delta/2$, and since $\Delta \leq \min(\log(1 + \epsilon/40), 1/2)$ is sufficient for Lemma 2, the interval width $\Delta$ does not depend on $n$. So, it remains to show that the range $T$ is $O(n)$.

At this step in the proof, we introduce notation related to the assumption that the eigenvalues of the covariances $\Sigma_j$ have constant upper and lower bounds. If the eigenvalues of $\Sigma_j$ are $\{\omega_i^j\}$, $i = 1, \ldots, n$, we require $\breve{\omega} \leq \omega_i^j \leq \hat{\omega}$, where $0 < \breve{\omega} \leq \hat{\omega}$ are the fixed bounds. Intuitively, upper bounds relate to the typical case in practice that input distributions have bounded variance. The lower bound on eigenvalues prevents the model from approaching degenerate Gaussian distributions, or equivalently having arbitrarily sharp spatial detail.

Our network is designed with $T = 4R^2 = 4\log(1/\tilde{\lambda}) = 4\log(\beta) + 4\log(1/\lambda)$. Recall from Section 1.4 that $\beta = \phi w(2\pi)^{-n/2}|\Sigma|^{-1/2}$ for fixed probabilities $\phi$ and $w$. From the assumed lower bound on eigenvalues of $\Sigma$, we have $|\Sigma|^{-1/2} \leq (1/\breve{\omega})^{n/2}$, and so $4\log(\beta)$ is $O(n)$. It remains to consider the scaling of $\log(1/\lambda)$ with $n$.

For a given probability $q > 0$, Lemma 3 in the main body requires that $\lambda = t^*\delta/(2J(1 + \delta))$, where $t^*$ is such that $\mathrm{P}[d(\mathbf{x}) < t^*] \leq q$, or equivalently, $\mathrm{P}[p(\mathbf{x}) < t^*/\phi] \leq q$, where $\phi$ is the fixed prior probability. Since $J$ and $\delta$ are fixed, we need to show that $\log(1/t^*)$ is $O(n)$. The existence of such a $t^*$ is established by Lemma 5. By assumption in Theorem 1, the variance of each Gaussian distribution is upperbounded by $\hat{\omega}$. Hence Lemma 5 shows that $t^*$ has the required scaling; specifically that as long as

$$\log(\phi/t^*) \geq \frac{n}{32}\log(32\pi\hat{\omega}) + \frac{1}{4}\log(1/q) + \log(J),$$

then $\mathrm{P}[p(\mathbf{x}) < t^*/\phi] \leq q$.

To finish the proof, we bound the growth rate of the coefficients used in the constructed two-hidden-layer network. Recall that the input coefficients of the first hidden layer are formed by the matrix $\Sigma_j^{-1/2}$ used to compute the values $y_i$, followed by the scaling $1/a$ in the input of the super-node functions $h(y_i, a)$. In addition to the lower bounds on $a$ required by Lemma 1, we can also require $a \geq 1$, for example, so that the first-layer coefficients are determined by the elements of the matrix $\Sigma_j^{-1/2}$. Since $\Sigma_j$ is a positive definite matrix, each element of $\Sigma_j^{-1/2}$ is bounded in magnitude by the trace, which in turn is upper bounded by $n\breve{\omega}^{-1/2}$, where $\breve{\omega}$ is the assumed lower bound on the eigenvalues of $\Sigma_j$. Hence the input coefficients of the first hidden layer grow no faster than $O(n)$.

The linear coefficients between the first and second hidden layers consist of the factor $a^2/\sigma^{(2)}(\tau)$ used in forming the first layer super-nodes $h(y_i, a)$, multiplied by the factor $1/\Delta$ used at the input to the supernodes $\psi(x, \sigma)$. The requirements on $\Delta$ in Lemma 2, and the constant $\sigma^{(2)}(\tau)$ are independent of $n$, and so it remains only to consider the scaling of the factor $a^2$. From the requirements of Lemma 3 in the main body, we have that $a$ must scale at least as fast as $R^3/\nu$, where $1/\nu$ is $O(n)$. We also have that $R^2 = \log(1/\tilde{\lambda})$, which was shown earlier to be $O(n)$. Putting these factors together, we have that $a$ is $O(n^{5/2})$ and the coefficients between first and second layer can be constructed as $O(n^5)$.

The linear coefficients in the output layer include the $o(1)$ coefficients $\exp(-k\Delta) - \exp(-(k - 1)\Delta)$ used to construct the super-node in (4), and the final multiplier $\beta_j$ in the expression $d(\mathbf{x}) = \sum_j \beta_j c_j(\mathbf{x})$. Similar to the scaling done for the single hidden layer network in Section 4, we can define the renormalized system $d'(\mathbf{x}) = \sum_j \beta_j' c_j'(\mathbf{x})$ where $\mathrm{E}[d'(\mathbf{x})^2] = 1$ and $\mathrm{E}[c_j'(\mathbf{x})^2] = 1$. Since all terms are positive, we have

$$1 = \mathrm{E}[d'(\mathbf{x})^2] \geq \sum_j (\beta_j')^2 \, \mathrm{E}[c_j'(\mathbf{x})^2] \geq \sum_j (\beta_j')^2,$$

showing that the last layer coefficients have a constant bound not depending on $n$.

In summary, we have shown that the coefficients used in the two-hidden-layer construction are polynomially bounded, in addition to the main result that the number of nodes is $O(n)$.

## 6    Proof of Theorem 2

Define $\mathbf{r} \in \mathbb{R}^{n_1}$, such that, for $i = 1, \ldots, n_1$,

$$r_i \triangleq \mathrm{E}\left[\mu_c(\mathbf{x})h_i(\langle \mathbf{w}_i, \mathbf{x} \rangle)\right].$$

The mean squared error can be lower-bounded as

$$
\begin{aligned}
\mathrm{E}[|\mu_c(\mathbf{x}) - f(\mathbf{x})|^2] &\geq& E[|\mu_c(\mathbf{x})|^2] - 2\,\mathrm{E}\left[\mu_c(\mathbf{x})f(\mathbf{x})\right] + E\left[|f(\mathbf{x})|^2\right] \\
&\geq& 1 - 2\sum_i a_i\,\mathrm{E}\left[\mu_c(\mathbf{x})h_i(\langle \mathbf{w}_i, \mathbf{x} \rangle)\right] \\
&\geq& 1 - 2\|\boldsymbol{a}\|\|\mathbf{r}\|,
\end{aligned}
$$

where the last step follows from the Cauchy-Schwarz inequality.

We next bound the value of $r_i$, showing that each is exponentially small in $n$. By the rotational symmetry of $\mu_c$ and $\mu$, we have

$$\mathrm{E}\left[\mu_c(\mathbf{x})h_i(\langle \mathbf{w}_i, \mathbf{x} \rangle)\right] = \mathrm{E}[\mu_c(\mathbf{x})h_i(x_1)].$$

Defining $\alpha = 1 + 2s_x/s_f$ and $\beta = (1/s_x + 1/s_f)^{-1}$, we write

$$
\begin{aligned}
r_i &=& \alpha^{n/4}\,(2\pi s_x)^{-n/2} \int h_i(x_1) e^{-\frac{1}{2\beta}\|\mathbf{x}\|^2} d\mathbf{x} \\
&=& \alpha^{n/4}\,(2\pi s_x)^{-n/2}\,(2\pi\beta)^{(n-1)/2} \int h_i(x_1) e^{-\frac{1}{2\beta}x_1^2} dx_1 \\
&=& \alpha^{n/4}\,(\beta/s_x)^{n/2} \int h_i(x_1) e^{-\frac{1}{2\beta}x_1^2} (2\pi\beta)^{-1/2}\, dx_1
\end{aligned}
$$

Note that $\int h_i(x_1) e^{-\frac{1}{2\beta}x_1^2} (2\pi\beta)^{-1/2}\, dx_1$ is the expected value of $h_i(x_1)$, with respect to $x_1 \sim \mathcal{N}(0, \beta)$. Therefore, using Jensen's inequality,

$$\left(\int h_i(x_1) e^{-\frac{1}{2\beta}x_1^2} (2\pi\beta)^{-1/2}\, dx_1\right)^2 \leq \int (h_i(x_1))^2 e^{-\frac{1}{2\beta}x_1^2} (2\pi\beta)^{-1/2}\, dx_1.$$

Therefore,

$$
\begin{aligned}
r_i^2 &\leq& \alpha^{n/2}\,(\beta/s_x)^n \int (h_i(x_1))^2\, e^{-\frac{1}{2\beta}x_1^2} (2\pi\beta)^{-1/2}\, dx_1 \\
&\leq& \alpha^{n/2}\,(\beta/s_x)^{n-1/2} \int (h_i(x_1))^2\, e^{-\frac{1}{2s_x}x_1^2} (2\pi s_x)^{-1/2}\, dx_1 \\
&\leq& \alpha^{n/2}\,(\beta/s_x)^{n-1/2}
\end{aligned}
$$

where the second step holds because $\beta = s_x s_f/(s_f + s_x) < s_x$, and therefore, $\exp(-x_1^2/(2\beta)) < \exp(-x_1^2/(2s_x))$, and the last step follows from our initial assumption that $\mathrm{E}[(h_i(x_1))^2] = 1$. Noting that

$$\alpha^{n/4}\,(\beta/s_x)^{n/2} = \left[\left(\frac{s_f + 2s_x}{s_f}\right)\left(\frac{s_f}{s_f + s_x}\right)^2\right]^{n/4} = \left(\frac{s_f^2 + 2s_f s_x + s_x^2}{s_f^2 + 2s_f s_x}\right)^{-n/4} = \rho^{-n/4},$$

we obtain

$$|r_i| \leq \rho^{-n/4}(1 + s_x/s_f)^{1/4} \tag{6}$$

This establishes that $\|\mathbf{r}\| \leq \sqrt{n_1}\rho^{-n/4}(1 + s_x/s_f)^{1/4}$, which finishes the proof.

# 7 Proof of Theorem 3

Consider random weights $\mathbf{w}_i \in \mathbb{R}^n$, $i = 1, \ldots, n_1$, that are mutually independent and distributed as $\mathcal{N}(\mathbf{0}, I_n)$. By construction, given weights $\mathbf{w}_1, \ldots, \mathbf{w}_{n_1}$, we have

$$f_{\mathbf{w}}(\mathbf{x}) = \frac{\alpha^{\frac{n}{4}}}{n_1} \sum_{i=1}^{n_1} \cos\left(\frac{1}{\sqrt{s_f}} \langle \mathbf{w}_i, \mathbf{x} \rangle\right). \tag{7}$$

Here the subscript $\mathbf{w}$ highlights the dependency of this function of the specific values of the weights. For a fixed $\mathbf{x}$, $\langle \mathbf{w}_i, \mathbf{x} \rangle$ is a zero-mean Gaussian random variable with variance $\|\mathbf{x}\|^2$. Therefore, since $\mathbf{w}_i$'s are i.i.d. themselves, for a fixed $\mathbf{x}$, $\cos(\frac{1}{s_f} \langle \mathbf{w}_i, \mathbf{x} \rangle)$, $i = 1, \ldots, n_1$, are i.i.d. bounded random variables. Moreover,

$$\mathrm{E}_{\mathbf{w}}\left[\cos\left(\frac{1}{\sqrt{s_f}} \langle \mathbf{w}_i, \mathbf{x} \rangle\right)\right] = \frac{1}{\sqrt{2\pi}} \int \cos\left(\frac{\|\mathbf{x}\|}{\sqrt{s_f}} u\right) \mathrm{e}^{-\frac{u^2}{2}} du$$

$$\overset{(a)}{=} \mathrm{e}^{-\frac{1}{2s_f}\|\mathbf{x}\|^2}, \tag{8}$$

where $(a)$ holds because of the following identity

$$\int \mathrm{e}^{-at^2} \cos(bt) dt = \sqrt{\frac{\pi}{a}} \mathrm{e}^{-\frac{1}{4a} b^2}.$$

Therefore, for a fixed $\mathbf{x}$,

$$\mathrm{E}_{\mathbf{w}}[f_{\mathbf{w}}(\mathbf{x})] = \alpha^{\frac{n}{4}} \mathrm{e}^{-\frac{1}{2s_f}\|\mathbf{x}\|^2} = \mu_c(\mathbf{x}). \tag{9}$$

The expected approximation error corresponding to a fixed $\mathbf{w}$ can be written as

$$\mathrm{E}_{\mathbf{x}}\left[(f_{\mathbf{w}}(\mathbf{x}) - \mathrm{E}_{\mathbf{w}}[f_{\mathbf{w}}(\mathbf{x})])^2\right]. \tag{10}$$

But, for a fixed $\mathbf{x}$,

$$\mathrm{E}_{\mathbf{w}}\left[(f_{\mathbf{w}}(\mathbf{x}) - \mathrm{E}_{\mathbf{w}}[f_{\mathbf{w}}(\mathbf{x})])^2\right] = \frac{\alpha^{\frac{n}{2}}}{n_1} \mathrm{var}_{\mathbf{w}}\left(\cos\left(\frac{1}{\sqrt{s_f}} \langle \mathbf{w}_i, \mathbf{x} \rangle\right)\right), \tag{11}$$

where

$$\mathrm{var}_{\mathbf{w}}\left(\cos\left(\frac{1}{\sqrt{s_f}} \langle \mathbf{w}_i, \mathbf{x} \rangle\right)\right) = \frac{1}{\sqrt{2\pi}} \int \cos^2\left(\frac{\|\mathbf{x}\|}{\sqrt{s_f}} u\right) \mathrm{e}^{-\frac{u^2}{2}} du - \mathrm{e}^{-\frac{1}{s_f}\|\mathbf{x}\|^2}$$

$$= \frac{1}{2\sqrt{2\pi}} \int \left(1 + \cos\left(\frac{2\|\mathbf{x}\|}{\sqrt{s_f}} u\right)\right) \mathrm{e}^{-\frac{u^2}{2}} du - \mathrm{e}^{-\frac{1}{s_f}\|\mathbf{x}\|^2}$$

$$= \frac{1}{2} + \frac{1}{2} \mathrm{e}^{-\frac{2}{s_f}\|\mathbf{x}\|^2} - \mathrm{e}^{-\frac{1}{s_f}\|\mathbf{x}\|^2}. \tag{12}$$

Therefore, from (11), we have

$$\mathrm{E}_{\mathbf{w}}\left[(f_{\mathbf{w}}(\mathbf{x}) - \mathrm{E}_{\mathbf{w}}[f_{\mathbf{w}}(\mathbf{x})])^2\right] = \frac{\alpha^{\frac{n}{2}}}{n_1} \left(\frac{1}{2} + \frac{1}{2} \mathrm{e}^{-\frac{2}{s_f}\|\mathbf{x}\|^2} - \mathrm{e}^{-\frac{1}{s_f}\|\mathbf{x}\|^2}\right). \tag{13}$$

Taking the expected value of both sides with respect of $\mathbf{x}$, and applying the Fubini's theorem (see e.g. [2]) to the left hand side, it follows that

$$\mathrm{E}_{\mathbf{w}}\left[\mathrm{E}_{\mathbf{x}}[[(f_{\mathbf{w}}(\mathbf{x}) - \mathrm{E}_{\mathbf{w}}[f_{\mathbf{w}}(\mathbf{x})])^2]]\right] = \frac{\alpha^{\frac{n}{2}}}{2n_1} + \frac{\alpha^{\frac{n}{2}}}{2n_1} \mathrm{E}_{\mathbf{x}}[\mathrm{e}^{-\frac{2}{s_f}\|\mathbf{x}\|^2}] - \frac{\alpha^{\frac{n}{2}}}{n_1} \mathrm{E}_{\mathbf{x}}[\mathrm{e}^{-\frac{1}{s_f}\|\mathbf{x}\|^2}]$$

$$= \frac{\alpha^{\frac{n}{2}}}{2n_1} \left(1 + \left(\frac{s_f}{s_f + 4s_x}\right)^{\frac{n}{2}} - 2\left(\frac{s_f}{s_f + 2s_x}\right)^{\frac{n}{2}}\right)$$

$$< \frac{\alpha^{\frac{n}{2}}}{n_1}. \tag{14}$$

But by assumption $n_1 > \frac{1}{\epsilon} \alpha^{\frac{n}{2}}$, therefore,

$$\mathrm{E}_{\mathbf{w}}\left[\mathrm{E}_{\mathbf{x}}[[(f_{\mathbf{w}}(\mathbf{x}) - \mathrm{E}_{\mathbf{w}}[f_{\mathbf{w}}(\mathbf{x})])^2]]\right] < \epsilon, \tag{15}$$

This shows that there exists at least one set of weights $\mathbf{w}_1, \ldots, \mathbf{w}_{n_1}$ which satisfies the desired error bound.