[Reviews · NeurIPS 2019]

Reviewer 1



Clarity: The paper is very well written and the contributions are clearly expressed. Quality: Technical quality of the presentation is high. Originality: the techniques seem to be original (although I am not an expert in this area, so my knowledge of the surrounding literature is somewhat limited). Significance: Unclear. The authors use the adjectives "shallow" and "deep" in several places but really they seem to be contrasting 1- vs. 2-hidden-layer networks. It is not clear how/whether the approach can lead to non-trivial separation results for *actual* deep networks. === Update after rebuttal period: bumping the score up, please add formal comparisons with Eldan-Shamir '16.

Reviewer 2



This paper takes a commonly used machine learning model, the Gaussian mixture model (GMM) whose discriminant function for classification task, as the target for approximation by neural networks, and provides clear theoretical analyses over it. To be specific, it proves the necessity and sufficiency for a two layer neural network with a linear number of nodes to approximate the GMM, and also the necessity of a single layer network with the exponential number of ndoes. The paper makes good contribution to consolidate the theoretical basis of the deep learning methods. === After rebuttal: no change of the score. Regarding the deep learning, the theoretical works are less than sufficient. This one counts for a piece.

Reviewer 3



In this article, the authors studied the approximation complexity of the discriminant function of GMMs, and showed that deeper networks (i.e., neural network models with two hidden layers) can exponentially reduce the number of parameters needed, compared to shallow ones with a single hidden layer. In this vein, I find this work complements the existing literature by considering the practical relevant context of separating GMMs. If the authors had manages to relax the Assumptions 1-3 to cover more practical settings, my rating would be higher, but I think it is entirely appropriate to postpone such improvements to future work. **After rebuttal**: I have read the author response and my score remains the same.

[Author Response · NeurIPS 2019]

We greatly appreciate all the reviewers' comments and feedback. Here are our responses.

## Reviewer 1

– *Significance: Unclear. The authors use the adjectives "shallow" and "deep" in several places but really they seem to be contrasting 1- vs. 2-hidden-layer networks. It is not clear how/whether the approach can lead to non-trivial separation results for \*actual\* deep networks.*

**A:** We agree with the reviewer that the end goal should be understanding the effectiveness of DNNs that have many layers, however, we believe that our results are steps in that direction.

In this work, we are using the terms "shallow" versus "deep" as is common in the deep learning literature. (For instance, one of our main references, (Eldan-Shamir 2016), also uses these terms exactly as defined in this paper.)

– *Consider including a more comprehensive comparison with existing separation results (e.g Eldan-Shamir 2016). For example, even though both show polynomial upper bounds in the size of the hidden layers, it appears that there exists a significant reduction in the degree of the polynomial in the current manuscript.*

**A:** This is a good point. In the revised version, we will add the following paragraph at the end of Section 7 which is on related work: "For a specialized radial function in $R^n$, [Eldan-Shamir 2016] shows that while any SNN would require at least exponentially many nodes to approximate the function, there exists a DNN with two hidden layers and $O(n^{19/4})$ nodes that well approximates the same function. In this paper, for a general class of widely-used functions viz GMM discriminant functions, we show that while SNNs require at least exponentially many nodes, for any GMM discriminant function there exists a DNN with two hidden layers and only $O(n)$ nodes that approximates it."

## Reviewer 2

– *No numerical results are provided to this paper. For a theoretical work, this can be acceptable. But some experimental results should be no harm.*

**A:** We agree that some numerical evaluations could be interesting for our achievability results. As a matter of due diligence, we did run numerical cross checks but decided to prioritize discussion of the theoretical results.

## Reviewer 3

– *If the authors had managed to relax the Assumptions 1-3 to cover more practical settings, my rating would be higher, but I think it is entirely appropriate to postpone such improvements to future work.*

**A:** First, please note that Assumptions 1-3 are only required for proving Theorem 1 (our achievability result). As explained in Section 4, Theorem 2 (i.e., our converse result) holds for general activation functions. In fact, it also applies to the cases where each node is allowed to have a different activation function. Second, as explained in the conclusion section, we could relax these assumptions, and allow e.g. ReLU activation function at a minor expense to the size of the network. In fact, we are currently working on various relaxations of these assumptions.

– *Consider changing the title: I personally find the word "approximation" misleading, the objective here is not to learn or to approximate GMMs, but instead to separate them;*

**A:** Even though we are motivated by classification of GMMs, we chose to put 'approximation' in the title, as our main technical results focus on approximating the discriminant functions of GMMs.

– *Clarify if the conclusion in Section 5 on the sufficiency of exponentially many nodes holds true if the covariance of x is \*not\* proportional to identity, and if its holds for a larger class of activation function sigma;*

**A:** We are able to show that the result holds for a larger class of activation functions (recent work) and we expect that it holds for more general covariance matrices as well (planned next step).

[Meta-Review · NeurIPS 2019]

The reviewers unanimously liked and recommended to accept the paper. The author feedback clarified some concerns that the reviewers had initially held.